# Higher level of acute serum VEGF and larger infarct volume are more frequently associated with post-stroke cognitive impairment

Astuti Prodjohardjono[1,2], Amelia Nur Vidyanti[2]*, Noor Alia Susianti[3], Sudarmanta[4], Sri Sutarni[2], Ismail Setyopranoto[2]

1 Doctorate Program of Medical and Health Science, Faculty of Medicine, Public Health, and Nursing, Universitas Gadjah Mada, Yogyakarta, Indonesia, 2 Department of Neurology, Faculty of Medicine, Public Health and Nursing, Universitas Gadjah Mada, Yogyakarta, Indonesia, 3 Neurology Research Office, Department of Neurology, Faculty of Medicine, Public Health and Nursing, Universitas Gadjah Mada, Yogyakarta, Indonesia, 4 Department of Radiology, Faculty of Medicine, Public Health and Nursing, Universitas Gadjah Mada, Yogyakarta, Indonesia

* amelia.nur.v@ugm.ac.id

**Data Availability Statement:** All relevant data are within the paper.

## Abstract

### Background

Serum vascular endothelial growth factor (VEGF) and infarct volume detected by brain imaging have been associated with stroke outcome. However, the relationship of these two variables with post-stroke cognitive impairment (PSCI) remains unclear. We aimed to investigate the association between acute serum VEGF levels and infarct volume with PSCI in ischemic stroke patients.

### Methods

Fifty-six first-ever ischemic stroke patients who were hospitalized in Dr. Sardjito General Hospital Yogyakarta, Indonesia were prospectively recruited. Serum VEGF level was taken on day 5 of stroke onset and measured by ELISA. Infarct volume was calculated manually from head CT scan by expert radiologist. PSCI was assessed after 3 months follow up by using Montreal Cognitive Assessment-Indonesian version (MoCA-INA). We performed a ROC curve analysis to determine the cut-off point of VEGF level and infarct volume. Multivariate logistic regression analysis was performed to measure the contribution of VEGF level and infarct volume to PSCI after controlling covariates (demographic and clinical data).

### Results

The mean age of PSCI and non-PSCI patients was 61.63% ± 8.47 years and 58.67% ± 9.01 years, respectively ($p = 0.221$). No differences observed for vascular risk factors, infarct location, and NIHSS in both groups. Multivariate logistic regression showed that patients with higher VEGF level alone ($\geq$519.8 pg/ml) were 4.99 times more likely to have PSCI than those with lower VEGF level (OR = 4.99, 95% CI = 1.01–24.7, $p = 0.048$). In addition, patients with larger infarct volume alone ($\geq$0.054 ml) were also more frequently associated with PSCI (OR = 7.71, 95% CI = 1.39–42.91, $p = 0.019$).

**Funding:** This study was fully supported by research grant from Dr. Sardjito General Hospital Yogyakarta, Indonesia (Grant number: H.K.02.03/XI.2/19020/2018). There was no additional external funding received for this study.

**Competing interests:** The authors have declared that no competing interests exist.

## Conclusions

Acute ischemic stroke patients with higher serum VEGF level (≥519.8 pg/ml) and larger infarct volume (≥0.054 ml) were more likely to have PSCI 3 months after stroke. These findings may contribute to predict PSCI earlier and thus better prevention strategy could be made.

## Introduction

Stroke survivors are susceptible to develop vascular cognitive impairment at 3 months after stroke [1]. This condition mostly described as post-stroke cognitive impairment (PSCI) [2–4]. The prevalence of PSCI ranges from 20% to 80% and varies between countries, races, and diagnostic criteria [2]. Based on the South London Stroke Register 1995–2010, the prevalence of PSCI was 22% and remained unchanged at 5 years after stroke [3]. Meanwhile, a multicenter prospective stroke cohort in Korea revealed the prevalence of PSCI was 62.6% [5].

In Indonesia, the prevalence of PSCI within 6 months of ischemic stroke is still high, reaching 68.2% [6]. Preventive, promotive, and curative efforts are needed to sustain the quality of life of PSCI patients, including to reduce the cognitive disability in daily living and prevent the progression to vascular dementia (VaD) or Alzheimer's disease (AD) or both [7]. The progression of PSCI to dementia would make the treatment more complicated and the prognosis worse [4, 7]. Hence, early detection and prediction of PSCI are essential to prevent this condition.

Vascular Endothelial Growth Factor (VEGF) is a dimeric glycoprotein with diverse effects, such as angiogenic and neuroprotective effects [8]. VEGF is one of the biomarkers that can be detected as early as 2–4 hours after the onset of stroke and may last for at least 28 days [9, 10]. VEGF is responsible for the process of angiogenesis after the incidence of ischemia [10]. VEGF reaches the highest level in serum on day 7 after stroke [11]. A study found that lower VEGF level on day 5 of stroke onset was associated with brain edema in acute ischemic stroke patients [12]. Although they did not directly measure the cognitive function, brain edema itself may cause cognitive dysfunction [12].

Neuroprotective effects of VEGF have been described in several *in vitro* studies by promoting neuronal survival [13–15]. This protective effect was also demonstrated in animal models of stroke. Intravenous administration of VEGF given at 48 hours after ischemia showed attenuation of neurobehavioral function in rats [16]. VEGF is upregulated particularly in the ischemic penumbra which surrounds the infarct core [17]. Administration of intraventricular VEGF to rats at 24 hours after ischemia and continued for 3 days showing a reduction in infarct volume by approximately one-third at 1 month post-stroke, as well as improvement of sensorimotor and cognitive impairment which persisted for at least 2 months [18, 19]. These studies suggest that VEGF may be beneficial for improving cognitive performance after ischemic stroke, either by reducing the infarct volume or promoting neuronal survival. This led us to hypothesize that VEGF levels may be altered in acute ischemic stroke which is associated with infarct volume and the development of PSCI at 3 months after stroke.

Larger infarct volume corresponds with cognitive impairment after stroke [20, 21]. Another study also found that brain infarcts were associated with a smaller hippocampus, and that both infarcts and reduced hippocampal volume were independently associated with memory decline [22]. However, some studies have shown no association between larger stroke and

cognitive impairment [23, 24]. Although contradictory, we believe that infarct volume contributes to the development of PSCI.

By understanding the importance of acute serum VEGF level and infarct volume in ischemic stroke patients, the development of PSCI may be predicted earlier thus better prevention strategy could be developed. In the present study, we aimed to investigate the association between serum VEGF level and infarct volume with the development of PSCI at 3 months after ischemic stroke.

## Materials and methods

### Study design and participants

This study was hospital-based with observational cohort. We recruited acute ischemic stroke patients which hospitalized at Neurological ward in Dr. Sardjito General Hospital Yogyakarta, Indonesia during June 2018- May 2019 who met the inclusion criteria. Patients who consented to the study were prospectively enrolled and followed up at 3 months after stroke onset. The inclusion criteria were: (1) Male and female patients who were hospitalized with first-ever stroke and full of consciousness (compos mentis); (2) Within day 5 of the stroke onset; (3) Age more than 18 year-old; (3) Having at least 3 years of educational experience at basic/elementary level; (4) Cooperative, can read and write; (5) Not taking memory enhancing drugs such as donepezil, galantamine, memantine, citicholine, piracetam, ginkgo biloba, folic acid, vitamin B, vitamin E, and selegiline. The exclusion criteria were: (1) Patients with history of central nervous system diseases such as: previous stroke, tumor, trauma, encephalitis, Parkinson's disease, epilepsy, and dementia or mild cognitive impairment, (2) Patients with other diseases that manifest in cognitive impairment, such as hepatitis, HIV, chronic alcohol use, and Wernicke-Korsakoff syndrome; (3) Patients with depression as screened by Hamilton Depression Rating Scale; (4) Aphasia or dysphasia; (5) Dysarthria; (6) Visual or hearing impairment; (7) Illiterate.

All patients had the following assessments: head CT scan immediately after patients' initial assessment in the emergency room, serum VEGF level and stroke severity at day 5 of stroke onset, and cognitive function to determine PSCI at 3 months after stroke onset. All the examiners were blinded and written informed consent was obtained from the participants.

From a total sample of 83 patients that were originally screened and met the inclusion criteria, 7 were lost to follow up, 1 died before the diagnosis of PSCI was established, and 19 were excluded from the analysis due to no lesion detected in the head CT scan. The final samples assessed at 3 months after stroke and included in the analysis was 56 patients (Fig 1).

### Data collection

**Demographic and clinical characteristics.**  This included age, sex, body mass index (BMI), education, the existence of vascular risk factors (comprised of hypertension, diabetes mellitus, smoking, dyslipidemia, and cardiovascular diseases), anti-dyslipidemic use, VEGF level, infarct volume, infarct location (region), infarct side (hemisphere), and stroke severity as measured by The National Institute of Health Stroke Scale (NIHSS). We did not include the data of acute stroke treatment due to all patients received similar standard treatment based on the stroke guideline. However, we included anti-dyslipidemic use as one of the variables because the drugs were only given to patients with dyslipidemia. NIHSS were only divided into 2 categories (minor and moderate) because there were no patients with categories of moderate to severe or severe. All demographic and clinical characteristics data were collected at the acute phase of ischemic stroke (during hospitalization).

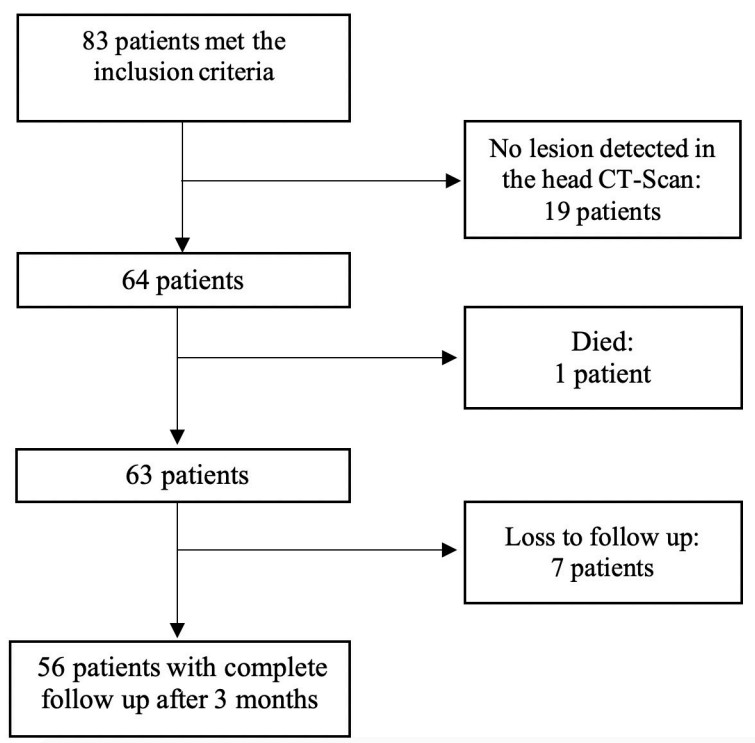

**Fig 1. Flow chart of patient selection.**

**Infarct volume.** Head CT scan was performed immediately after patients' assessment in the emergency room. Philips Ingenuity 128 MSCT machine was used in this study, with scanning parameter was peaked at 120 kVp, table speed 52.9 mm/second (pitch, 0.828–1.07), section scan time 9.3 seconds. Slide thickness was at 0,625 mm during image acquisition and 1 mm during image reconstruction which was displayed on the workstation's software. Isotropic 1x1x1 mm$^3$ voxel enables 3D-multiplanar reconstruction with native resolution. Philips IntelliSpace Portal version 10.1 software were used for hypodense area segmentation. Segmentation module and volume rendering technique were used to obtain 3D visualization of the lesion in each segmentation. Infarct volume calculation was done manually using hand tracing method at the edge of hypodense area by expert radiologist specialize in brain imaging. Edge of lesion segmentation was carefully examined, slide by slide, until the last slide in which the minimum hypodense lesion area was still visualized. The segmentation process was done axially which then confirmed by the coronal and sagittal planes.

Stroke window method was used for ischemic area visualization, by adjusting window width at 8 HU and window center at 32 HU. Ischemic area is defined by parenchymal density less than 20 HU. Cerebrospinal fluid or old infarct that degenerates into encephalomalacia has a density less than 5 HU. CT scan images containing artifacts from patient's movement during the procedures that could affect segmentation process which led to inaccurate volume interpretation were excluded.

**Measurement of serum VEGF level.** Serum samples were taken at day 5 of stroke onset. A needle vacuum blood collection-21 (BD Vacutainer) was inserted into the antecubital vein and blood samples were taken between 7 until 9 a.m. after 10–12 hour fasting using vacuum blood collecting tubes (SST tubes BD Vacutainer). Then, serum samples were extracted and

stored at -20˚C until assays. Serum VEGF levels were determined by using a solid phase enzyme-linked immunosorbent assay (ELISA) kit specific for the quantitative determination of human VEGF[165] levels cell culture supernates, serum, and plasma (Quantikine®, R&D Systems, Inc., Minneapolis, MN, USA, Cat No: DVE00). The minimum detection limit of the assay was <9.0 pg/ml, and the intra- and inter-assay coefficients of variation were <10%. All of the assays were conducted in Prodia Laboratory Research, Jakarta.

**Analysis of VEGF.** A monoclonal antibody specific for human VEGF had been pre-coated onto 96 well polystyrene microplate (VEGF[165], R&D System, Cat No: MAB3045-SP) at concentration 1000 pg/mL in a buffered protein base with preservatives; lyophilized. The test samples were applied (100μL/well) in duplicate. A standard curve was obtained, using 200 μL/well of VEGF[165] (R&D systems) at concentrations of 0, 31.3, 62.5, 125, 250, 500, 1000, 2000 pg/mL, in duplicate wells. Each well was aspirated and washed, and the process was repeated twice for a total of three washes. Each well was washed by washing buffer (400 μL) using a squirt bottle, manifold dispenser, or autowasher. A 200 μL of human VEGF conjugate was added to each well. Subsequently, each well was covered with a new adhesive strip and incubated for 2 hours at room temperature. After that, we repeated the aspiration/wash procedure and added 200 μL of substrate solution to each well. We then incubated each well for 25 minutes at room temperature, followed by adding 50 μL of stop solution (2N Sulfuric acid) to each well. We determined the optical density of each well within 30 minutes, using a microplate reader set to 450 nm.

**Cognitive assessments.** We assessed the cognitive function at 3 months after stroke onset using Montreal Cognitive Assessment-Indonesian version (MoCA-INA). MoCA-INA is a 30-point cognitive screening tool which has been validated for Indonesia population. It has been widely accepted and commonly used in the clinical and community setting in Indonesia [25, 26]. The cut-off point to be assessed as cognitively impaired is <26 [27]. Patients with inability to use the dominant arm were told to use the non-dominant arm for performing the test. Scoring of MoCA-INA is based on a guideline issued by Indonesian Neurological Association. The guideline includes how the test is administered and scored. Each test item has its own criteria for scoring purposes [28]. In addition to MoCA-INA, at 3 months after stroke we also screened the level of depression for each patient using Hamilton Depression Rating Scale (HDRS). Patients with HDRS score >7 were excluded due to higher possibility to have post-stroke depression [29], which can affect the cognitive function. Patients with MoCA-INA score <26 and HDRS <7 were categorized as PSCI group, while those with MoCA-INA score ≥26 and HDRS <7 were non-PSCI group.

**Ethical considerations.** All procedures performed in the present study were in accordance with ethical standards of the institutional research committees and with the 2013 Declaration of Helsinki. Ethical approval for this study was obtained from the Medical and Health Research Ethics Committee of the Faculty of Medicine, Universitas Gadjah Mada, Indonesia (EC No. KE/FK/1100/EC/2018).

## Statistical analysis

To analyze the statistical difference between variables, we used independent t-test (for continuous variables), Mann-Whitney (for variables not normally distributed), and Chi square test (for categorical variables). To determine the cut-off point of VEGF level and infarct volume, we performed a receiver operating characteristic (ROC) curve analysis. The crude odds ratios of VEGF level and infarct volume associated with PSCI were measured using bivariate analysis with categorization from the cut-off point derived by the ROC curve. Finally, multivariate logistic regression analysis was performed to measure the contribution of VEGF level and

infarct volume to PSCI after controlling covariates such as infarct location (regions), infarct size (hemisphere) and NIHSS for Model 1; plus demographic characteristics for Model 2; plus vascular risk factors and anti-dyslipidemic use for Model 3. Interaction analysis of VEGF and infarct volume as well as VEGF and NIHSS were performed to observe the influence of infarct volume and stroke severity on the main effect of VEGF and PSCI. The association of VEGF level and cognitive function (subdomain MoCA score) was tested using general linear mixed model controlling the age, sex, education, infarct volume, and NIHSS. All statistical analyses were assessed by SPSS software version 25.0 (IBM Co. Ltd, NY, USA). A $p$ value of $<0.05$ in two-tailed test indicated statistical significance.

## Results

### Demographic and clinical characteristics

Table 1 presents the baseline characteristics of the patients. Thirty-five patients developed PSCI and 21 did not have PSCI. The mean age of patients with and without PSCI was 61.63% ± 8.47 years and 58.67% ± 9.01 years, respectively ($p$ = 0.221). There were no differences in

**Table 1. Baseline characteristics of all PSCI and non-PSCI patients after ischemic stroke.**

| Characteristics | PSCI | Non-PSCI | $p$ value |
|---|---|---|---|
| Age–year, mean ± SD | 61.63 ± 8.47 | 58.67 ± 9.01 | 0.221 |
| Sex–no. % | | | |
| Male | 24 (68.6%) | 11 (31.4%) | 0.226 |
| Female | 11 (52.4%) | 10 (47.6%) | |
| BMI–kg/m2, mean ± SD | 23.94 ± 3.29 | 24.63 ± 3.95 | 0.483 |
| Education–year, median (min-max) | 12 (4–16) | 12 (6–22) | 0.058 |
| Vascular risk factors–no. % | | | |
| Yes | 29 (67.4%) | 14 (32.6%) | 0.164 |
| No | 6 (46.1%) | 7 (53.9%) | |
| Anti-dyslipidemic use–no. % | | | |
| Yes | 27 (62.8%) | 16 (37.2%) | 1 |
| No | 8 (61.6%) | 5 (38.4%) | |
| VEGF–pg/ml, median (min-max) | 501.6 (70.3–1647.3) | 386.8 (149.2–1743.5) | 0.106 |
| Infarct volume–ml, median (min-max) | 0.19 (0.02–16.08) | 0.04 (0.003–1.49) | 0.003** |
| Infarct location (regions)–no. % | | | |
| Cortical | 3 (75.0%) | 1 (25.0%) | 0.317 |
| Subcortical | 29 (59.1%) | 20 (39.2%) | |
| Cortical and subcortical (both) | 3 (100.0%) | 0 (0.0%) | |
| Infarct side (hemisphere)–no. % | | | |
| Right | 12 (46.2%) | 14 (53.8%) | 0.018* |
| Left | 15 (68.2%) | 7 (31.8%) | |
| Right and left (both) | 8 (100.0%) | 0 (0.0%) | |
| NIHSS–no. % | | | 0.783 |
| Minor | 17 (60.7%) | 11 (39.3%) | |
| Moderate | 18 (64.3%) | 10 (35.7%) | |

PSCI, post-stroke cognitive impairment; BMI, body mass index; VEGF, vascular endothelial growth factor; NIHSS, The National Institutes of Health Stroke Scale; SD, standard deviation.

*$p$ <0.05;

**$p$ <0.01.

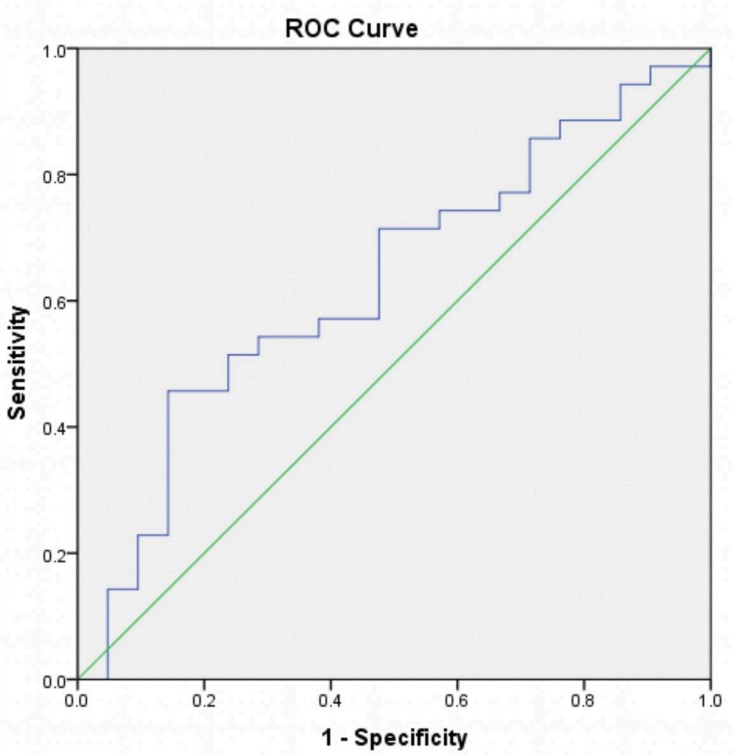

**Fig 2. ROC curve for the discrimination quality of VEGF level in PSCI patients (weak categorization).** Serum level of VEGF at cut-off point of 519.8 pg/ml with AUC 0.630 were used.

vascular risk factors, infarct location, and NIHSS for both groups. PSCI patients had larger infarct volume than those without PSCI (0.19 vs 0.04, $p$ = 0.003).

## Categorization of serum VEGF level and infarct volume

To determine the cut-off point of serum VEGF level and infarct volume in discriminating between PSCI and non-PSCI patients, we performed ROC analysis. We found that the cut-off point for serum VEGF level was 519.8 pg/ml with the AUC 0.630 (Fig 2). Meanwhile, the cut-off point for infarct volume was 0.054 ml with the AUC 0.739 (Fig 3). By using these cut-off points, we subsequently analyzed the association between VEGF level and infarct volume with PSCI in bivariate and multivariate analysis.

Table 2 shows bivariate analysis between PSCI with the categorization of VEGF level and infarct volume. Patients with higher VEGF level ($\geq$519.8 pg/ml) were more likely to have PSCI than those with lower VEGF level (<519.8 pg/ml) (OR = 5.05, 95% CI = 1.26–20.32, $p$ = 0.016). Patients with larger infarct volume ($\geq$0.054 ml) were also more likely to have PSCI than the counterpart group (<0.054 ml) (OR = 9.75, 95% CI = 2.67–35.53, $p$ <0.001).

## Factors associated with post-stroke cognitive impairment

To investigate whether serum VEGF level and infarct volume were independently associated with PSCI and could serve as predictors for developing PSCI, we performed a multivariate logistic regression analysis. Table 3 presents the multivariate logistic regression analyses of factors associated with PSCI after adjustment by controlling multiple covariates. In Model I,

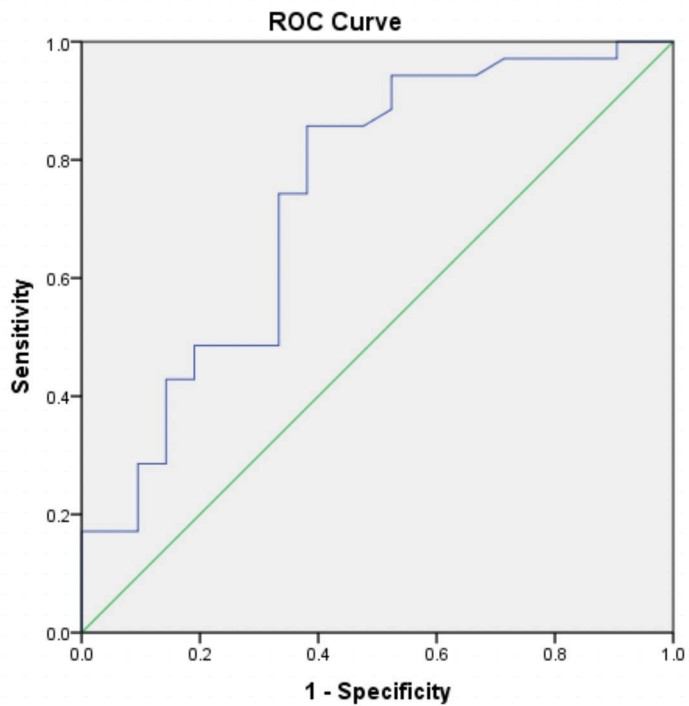

**Fig 3. ROC curve for the discrimination quality of infarct volume in PSCI patients (moderate categorization).**
Infarct volume at cut-off point of 0.054 ml with AUC 0.739 were used.

infarct volume was the only factor associated with PSCI in which larger lesion volume ($\geq$0.054) was more frequently associated with PSCI (OR = 5.57, 95% CI = 1.28–24.16, $p$ = 0.021). In Model II, we controlled the demographic characteristics and showed that in addition to larger infarct volume, patients with higher VEGF level ($\geq$519.8) were also more likely to develop PSCI (OR = 5.36, 95%CI = 1.35–21.41, $p$ = 0.017; OR = 4.66, 95% CI = 1.12–19.42, $p$ = 0.034, respectively).

For Model III, we adjusted the odds ratio of VEGF level and infarct volume by controlling all variables (clinical and demographic characteristics). We found that patients with higher VEGF level alone ($\geq$519.8 pg/ml) were 4.99 times more likely to have PSCI than those with

**Table 2. Bivariate analysis between PSCI with the categorization of VEGF level and infarct volume.**

| Variables | PSCI | | Non-PSCI | | OR | CI 95% | $p$ value |
|---|---|---|---|---|---|---|---|
| | n | % | n | % | | | |
| VEGF level | | | | | | | |
| $\geq$519.8 | 16 | 45.7% | 3 | 14.3% | 5.05 | 1.26–20.32 | 0.016* |
| <519.8 | 19 | 54.3% | 18 | 85.7% | Ref | | |
| Infarct volume | | | | | | | |
| $\geq$0.054 | 30 | 85.7% | 8 | 38.1% | 9.75 | 2.67–35.53 | <0.001** |
| <0.054 | 5 | 14.3% | 13 | 61.9% | Ref | | |

PSCI, post-stroke cognitive impairment; OR, odds ratio; CI, confidence interval; VEGF, vascular endothelial growth factor;

*$p$ <0.05;

**$p$ <0.01.

**Table 3. Multivariate logistic regression of factors associated with PSCI after adjustment for covariates.**

| Variable (Risk vs. Reference) | OR | CI 95% | p |
|---|---|---|---|
| **Model I** | | | |
| VEGF (≥519.8 vs. <519.8) | 2.32 | 0.54–9.83 | 0.242 |
| Infarct volume (≥0.054 vs. <0.054) | 5.57 | 1.28–24.16 | 0.021* |
| Infarct location (regions) | | | |
| (subcortical vs. cortical) | 0.89 | 0.05–13.68 | 0.963 |
| (both vs. cortical) | ∞ | 0.00 –∞ | 0.963 |
| Infarct side (hemisphere) | | | |
| (left vs. right) | 3.17 | 0.76–13.05 | 0.952 |
| (both vs. right) | ∞ | 0.00 –∞ | 0.943 |
| NIHSS (Moderate vs. Minor) | 0.77 | 0.19–3.15 | 0.845 |
| **Model II** | | | |
| VEGF (≥519.8 vs. <519.8) | 4.66 | 1.12–19.42 | 0.034* |
| Infarct volume (≥0.054 vs. <0.054) | 5.36 | 1.35–21.41 | 0.017* |
| Infarct location (regions) | | | |
| (subcortical vs. cortical) | 0.11 | 0.01–12.03 | 0.977 |
| (both vs. cortical) | 0.00 | 0.00 –∞ | 0.984 |
| Infarct side (hemisphere) | | | |
| (left vs. right) | 7.32 | 0.67–79.52 | 0.958 |
| (both vs. right) | 0.00 | 0.00 –∞ | 0.946 |
| NIHSS (Moderate vs. Minor) | 1.44 | 0.42–4.97 | 0.845 |
| Age (year) | 0.12 | 0.03–0.62 | 0.310 |
| Sex (female vs male) | 1.02 | 0.78–1.33 | 0.011* |
| BMI (kg/m$^2$) | 0.52 | 0.30–0.91 | 0.941 |
| Education (year) | 0.39 | 0.15–1.02 | 0.020* |
| **Model III** | | | |
| VEGF (≥519.8 vs. <519.8) | 4.99 | 1.01–24.7 | 0.048* |
| Infarct volume (≥0.054 vs. <0.054) | 7.71 | 1.39–42.91 | 0.019* |
| Infarct location (regions) | | | |
| (subcortical vs. cortical) | 0.26 | 0.01 –∞ | 0.991 |
| (both vs. cortical) | 0.04 | 0.00 –∞ | 0.979 |
| Infarct side (hemisphere) | | | |
| (left vs. right) | 12.13 | 1.39–42.91 | 0.959 |
| (both vs. right) | ∞ | 0.00 –∞ | 0.955 |
| NIHSS (Moderate vs. Minor) | 3.04 | 0.6–15.38 | 0.179 |
| Age (years) | 1.09 | 0.96–1.26 | 0.211 |
| Sex (male vs female) | 0.01 | 0.01–0.32 | 0.012* |
| BMI (kg/m$^2$) | 0.96 | 0.71–1.3 | 0.788 |
| Education (year) | 0.42 | 0.21–0.84 | 0.013* |
| Vascular risk factors | 0.45 | 0.09–2.28 | 0.328 |
| Anti-dyslipidemic use | 0.34 | 0.07–1.55 | 0.164 |
| **Interaction analysis** | | | |
| (VEGF x infarct volume) | 1.27 | 0.23–6.95 | 0.788 |
| (VEGF x NIHSS) | 0.78 | 0.15–4.01 | 0.757 |

PSCI, post-stroke cognitive impairment; OR, odds ratio; CI, confidence interval; BMI, body mass index; VEGF, vascular endothelial growth factor; NIHSS, The National Institutes of Health Stroke Scale;
*p <0,05.

**Table 4. Association between VEGF level (≥519.8 pg/ml) with cognitive function based on subdomain of MoCA-INA after adjustment of age, sex, education, infarct volume, and NIHSS.**

| MoCA-INA subdomain | Total sample (n = 56) | | | VCI patient (n = 35) | | | Non-VCI (n = 21) | | |
|---|---|---|---|---|---|---|---|---|---|
| | β | SE | p-value | β | SE | p-value | β | SE | p-value |
| Visuospatial | -0.646 | [0.3766] | 0.093 | -3.287 | [1.5171] | 0.042* | -0.927 | [1056.35] | 0.999 |
| Naming | -14.14 | [498.79] | 0.977 | -4.787 | [324.77] | 0.988 | -17.175 | [5290.06] | 0.997 |
| Attention | 0.407 | [0.3306] | 0.225 | -0.172 | [0.5753] | 0.767 | -45.045 | [5225.42] | 0.993 |
| Language | 0.245 | [0.5707] | 0.669 | 1.657 | [1.2143] | 0.187 | -43.313 | [1569.23] | 0.978 |
| Abstraction | 0.461 | [0.6377] | 0.473 | 1.766 | [1.2302] | 0.166 | 0.277 | [4068.63] | 0.999 |
| Recall | -0.332 | [0.2319] | 0.159 | -1.642 | [0.7032] | 0.030* | 29.337 | [1967.07] | 0.988 |
| Memory | 0.071 | [0.3773] | 0.849 | 0.379 | [0.7721] | 0.047 | 3.5777 | [1663.34] | 0.988 |

VEGF, Vascular Endothelial Growth Factor; MoCA–INA, Montreal Cognitive Assessment–Indonesian Version; VCI, Vascular Cognitive Impairment; SE, Standard Error;

*p<0.05.

lower VEGF level (OR = 4.99, 95% CI = 1.01–24.7, $p$ = 0.048). In addition, patients with larger infarct volume alone (≥0.054 ml) were also more frequently associated with PSCI (OR = 7.71, 95% CI = 1.39–42.91, $p$ = 0.019). Interaction analysis of VEGF and infarct volume with PSCI, as well as VEGF and NIHSS with PSCI revealed no significant interaction effect.

### Association between VEGF level with cognitive function

Table 4 shows the association between higher VEGF level (≥519.8 pg/ml) with cognitive function based on subdomain of MoCA-INA for each group of patients. Higher VEGF level (≥519.8 pg/ml) was significantly associated with deficits in visuospatial and recall domains for PSCI patient. PSCI patients with higher VEGF level had visuospatial score 3.287 lower and recall score 1.642 lower than the counterpart group. However, no significant association observed in non-PSCI patients.

### Discussion

In the present study, we demonstrated that PSCI was affected by acute serum VEGF level and infarct volume. Patients with higher VEGF level (≥519.8 pg/ml) and larger infarct volume (≥0.054 ml) were more likely to develop PSCI. To our knowledge, this is the first study which demonstrated that higher serum VEGF level in the acute ischemic stroke patients contribute to the development of PSCI.

The higher level of serum VEGF in PSCI patients in the present study was unexpected. Prior studies showed that decreased serum VEGF was positively correlated with a higher risk of Alzheimer's disease (AD) [30, 31]. Moreover, higher VEGF level had a protective effect and was associated with larger hippocampal volume, less hippocampal atrophy, and less cognitive decline in AD patients [32]. These studies support VEGF as a neuroprotectant that contributes to improvement of cognitive function. Nevertheless, some studies revealed that higher level of serum VEGF [33, 34] were found in AD patients which corresponded with the disease progression and severity of cognitive impairment. VEGF elevations may be due to a compensatory mechanism to prevent the clinicopathologic manifestations of AD [34], or a result from hypoxia/hypoperfusion state related to the vascular pathology of AD [35–37].

Contradictory to our findings, prior study using *in vitro* and *in vivo* experiment of stroke model demonstrated that VEGF administration improved cognitive impairment through alleviating neuronal function and viability [38]. Furthermore, other study revealed that serum

VEGF expression measured on day 5 of stroke onset was significantly lower in ischemic stroke patients than in control group [39]. Moreover, the increase of serum VEGF levels within 24 h of stroke onset was proportional to an improved NIHSS after 3 months [8]. Taken together, those prior studies depicted that higher VEGF levels would be beneficial in acute stroke in terms of cognitive function and clinical outcome.

On the other hand, our present study is in agreement with a prior study which reported that VEGF was highly upregulated in acute ischemic stroke patients with severe disability measured by higher NIHSS score [40]. Although they did not measure cognitive function as the main outcome, the results demonstrated that higher acute serum VEGF level (within 2 weeks of stroke onset) was not beneficial for long-term outcome in ischemic stroke patients. In addition, previous reports showed that higher serum VEGF level was correlated with infarct volume and stroke severity [8, 11]. However, in the present study we found no significant interaction effect between VEGF level with infarct volume and NIHSS. It depicts that the contribution of higher VEGF level to PSCI was not related to infarct volume. This further supports that higher serum VEGF was independently associated with higher possibility of developing PSCI.

The reason why VEGF level in our PSCI patients was higher remains elusive. There are some possible explanations for this phenomenon. First, VEGF serum in this study was taken at day 5 of ischemic stroke onset in which during that time the VEGF reaches the highest level in serum after a stroke attack [11]. Neurons, microglia, and astrocytes produced higher levels of VEGF in the penumbra surrounding infarct tissue in the patients with acute ischemic stroke [41, 42]. The upregulation of VEGF protein is a response triggered by hypoxic conditions associated with the ischemic process [43, 44]. The higher VEGF level in PSCI patients in the present study showed the possibility that the process of hypoxia/ischemia in PSCI patients were more robust and more severe than the counterpart group. Subsequently, it may trigger chronic cerebral hypoperfusion state, leading to cognitive impairment [45]. Second, the higher acute VEGF level has detrimental effects which counteract the beneficial actions by triggering vascular leakage and neuroinflammation due to increased permeability of blood-brain barrier and tissue damage after stroke [10, 19]. This condition may potentially cause cognitive impairment.

Infarct volume has been known as a significant risk factor for cognitive impairment following ischemic stroke [46–48]. In the present study, we found that larger infarct volume detected by the head CT scan was more frequently associated with PSCI, thus in accordance with previous studies. However, prior studies used MRI modality for assessing the ischemic/infarct volume because they recruited post-ischemic stroke patients (3 weeks-6 months after stroke onset). Therefore, MRI was chosen due to its superiority to detect ischemic lesion in subacute or chronic stroke [49, 50]. Nonetheless, MRI is more cost-effective and time consuming than CT scan. Hence, CT scan is still recommended as a gold standard of brain imaging which routinely used for all ischemic stroke patients [51].

Nevertheless, some studies demonstrated a contradictory finding that larger infarct volume was not associated with cognitive impairment [24, 52]. This discrepancy may be caused by different methodology used or the fact that size and location of infarct are confounding variables in stroke which could influence the cognitive function [20]. Regardless the different findings, larger infarct volume in our present study was independently associated with PSCI after adjusted by ischemic/infarct location and other potential confounding factors.

Prior studies have established that people with vascular risk factors are more likely to have PSCI [53–55]. However, we did not find any association between vascular risk factors with PSCI in the present study. This may due to the limited number of participants. The reason for this was because we collected the participants in a tertiary hospital with first-ever stroke with

the stroke onset was less than 5 days. Most of stroke patients in our tertiary hospital are recurrent stroke or with some complications (such as with loss of consciousness, established dementia). The second explanation may be related to the time period for following the development of cognitive impairment. Three months after stroke onset may be not sufficient to observe the impact of vascular risk factors to the development of PSCI.

We also found that higher VEGF level in PSCI patients was associated with deficits in visuospatial and recall domains. Vascular cognitive impairment related to stroke has been shown to be associated with disruption of fronto-cortical connections, including deficits in executive function, speed, and attention [56, 57]. However, cognitive deficits associated with PSCI can vary with impairment in global functioning, memory, or non-memory domain. The function of cognitive domain affected by stroke may depend on the size and location of the infarct [20, 57]. No prior study reported the association of VEGF level with the function of cognitive domain in PSCI. Hence, the possible explanation for our finding warrants further investigation.

The present study contributes to providing further evidence for a contribution of acute serum VEGF to predict PSCI 3 months after stroke. However, there are several limitations in this study. First, this was a single-center study. We only included participants from tertiary hospital in Yogyakarta, Indonesia which led to a small sample size. Therefore, the findings in the present study needs careful interpretation upon generalization. Second, serum VEGF was only measured during the acute phase of ischemic stroke owing to limited funding. The measurement during acute and chronic phase (3 months after stroke) may serve better relationship with the outcome of PSCI. Furthermore, this study covers a follow up period of only 3 months after stroke onset for determining the association of VEGF and PSCI. Follow-up for longer period is likely to strengthen the apparent difference in long-term risk of higher VEGF level between 2 group of patients with and without PSCI. Third, due to lack of feasibility and limited funding, we did not perform MRI at 3 months after stroke to evaluate the lesion volume and location in PSCI patients. In addition, the time of CT scan still varied between 0 and 5 days from stroke onset. The characteristics of the brain infarct on CT scan evolve overtime in the acute and subacute phase of stroke, since the lesion progressively acquires better contour definition and hypodensity, as well as for the development and resolution of cerebral edema. Hence, this could affect the accurate assessment of infarct volume. Finally, we did not investigate the detrimental mechanism of VEGF related to PSCI after stroke. It is known that robust expression of VEGF could lead to increased vascular permeability and neuroinflammation. Future study investigating the detailed signaling pathway of VEGF leading to PSCI in acute ischemic stroke patients might be beneficial.

## Conclusions

Acute ischemic stroke patients with higher serum VEGF level ($\geq$519.8 pg/ml) and larger infarct volume ($\geq$0.054 ml) were more likely to have PSCI at 3 months after stroke onset. These findings may contribute to predict PSCI earlier thus better prevention strategy could be made.

## Supporting information

**S1 Dataset. VEGF dataset.**
(XLSX)

## Acknowledgments

The authors are indebted to all the participants in this study for their commitment and cooperation. The authors also thank Muhammad Sari'uddin, Rangga Adi Nugraha, Putri Andhini, neurology residents, for their technical supports during the study; and staff at Prodia Laboratory Research, Jakarta for their technological assistance.

## Author Contributions

**Conceptualization:** Astuti Prodjohardjono, Amelia Nur Vidyanti, Sri Sutarni, Ismail Setyopranoto.

**Data curation:** Astuti Prodjohardjono, Noor Alia Susianti, Sudarmanta.

**Formal analysis:** Sudarmanta.

**Funding acquisition:** Astuti Prodjohardjono.

**Investigation:** Astuti Prodjohardjono.

**Methodology:** Astuti Prodjohardjono, Amelia Nur Vidyanti, Sri Sutarni, Ismail Setyopranoto.

**Project administration:** Noor Alia Susianti.

**Resources:** Astuti Prodjohardjono, Noor Alia Susianti.

**Software:** Astuti Prodjohardjono, Sudarmanta.

**Supervision:** Sri Sutarni, Ismail Setyopranoto.

**Validation:** Amelia Nur Vidyanti, Sri Sutarni, Ismail Setyopranoto.

**Visualization:** Amelia Nur Vidyanti.

**Writing – original draft:** Astuti Prodjohardjono, Amelia Nur Vidyanti.

**Writing – review & editing:** Amelia Nur Vidyanti, Sri Sutarni, Ismail Setyopranoto.

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
