## [Decision Letter · Decision Letter 0]

7 Jul 2020

PONE-D-20-13907

Higher level of acute serum VEGF and larger infarct volume are associated with higher risk of vascular cognitive impairment after ischemic stroke

PLOS ONE

Dear Dr. Vidyanti,

Thank you for submitting your manuscript to PLOS ONE. After careful consideration, we feel that it has merit but does not fully meet PLOS ONE’s publication criteria as it currently stands. Therefore, we invite you to submit a revised version of the manuscript that addresses the points raised during the review process.

We look forward to receiving your revised manuscript.

Kind regards,

Marietta Zille, PhD

Academic Editor

PLOS ONE

Journal Requirements:

2. Thank you for including your funding statement; "This study was supported in part by research grant from Dr. Sardjito General Hospital Yogyakarta, Indonesia (Grant number: H.K.02.03/XI.2/19020/2018)."

Additional Editor Comments (if provided):

Reviewer 1:

Dear authors,

The major concern is that the severity of neurological impairment (National Institute of Health Stroke Scale) was not assessed and may be very different between groups with and without Vascular Cognitive Impairment (VCI and non-VCI). Therefore, the higher levels of Vascular Endothelial Growth Factor may not be associated exclusively with ICV. Another concern is whether or not the cognitive state previous to the study was accessed. The Montreal Cognitive Assessment-Indonesia version (MoCA-INA) score on day 5 is unclear for both groups. These are the main remarks that should be discussed.

Further notes:

- Page three, line fifty-six: “In Indonesia, the prevalence of VCI after ischemic stroke is still high, reaching 68.2%”. It is necessary to add the ictus time (The ischemic stroke within 6 months).

- Page three, line sixty-seven: “Another study found that the level of VEGF on day 5 of stroke onset could predict the outcome in stroke patients”. Is the level of VEGF higher or lower?

- Were there patients with temporal lobe and/or hippocampus injuries? Such discussion has not been elaborated in the article.

- Page sixteen, line two hundred and seventy-four: “Another study showed lower serum VEGF expression was found in ischemic stroke patients, but not in control group”. It is also important to mention the ictus time here.

Reviewer 2:

In the manuscript entitled “Higher level of acute serum VEGF and larger infarct volume are associated with higher risk of vascular cognitive impairment after ischemic stroke” the Authors examined a population of 56 ischemic stroke patients and investigated the possible associations of serum levels of the Vascular Endothelial Grow Factor (VEGF) and ischemic lesion volume with vascular cognitive impairment at 3 months from stroke onset.

VEGF indeed is an important growth factor mediating several protective but also some detrimental effect in the ischemic brain.

Following aspects should be considered to improve the manuscript:

The possible inter-relationship between the two study variables (serum VEGF and lesion size) has not been investigated, so that the two variable appear poorly integrated and the manuscript loses cohesiveness.

- endpoint of the study: The endpoint of the study is the development of vascular cognitive impairment at 3 months from stroke onset. The Authors evaluated the development of vascular cognitive impairment by means of the dichotomized Montreal Cognitive Assessment (MoCA) scale (with a cut-off of 26).

- Since all included patients suffered from ischemic stroke, the term “post-stroke cognitive impairment” instead of “vascular cognitive impairment” appears more accurate and is suggested to the Authors (https://doi.org/10.1186/s12916-017-0779-7).

- Various tools are available to assess cognition after stroke, with clear standards missing, and need to be discussed. Impact on the activities of daily living, perceived impairment, as well as subdomain scores of the MoCA have not been investigated, but would provide additional information.

- Importantly, it has not been clarified how the test has been administered in cases of stroke patients with aphasia, dysphasia, dysarthria, hearing impairment, inability to use the dominant arm, or visual impairment.

- Similarly, stroke severity, lesion side (right/left), acute stroke treatment are easily available data with impact on stroke outcome that should have been included in patient description, as well as their association with cognitive performances should have been explored.

- Whereas the Authors stated that patients with history of dementia have been excluded from the sample, it is fundamental to clarify how it was assessed: e.g. by reports of an established diagnosis, screening questionnaire administered to the patient or to a family member, etc. An easy assessment of pre-stroke functional status, such as modified Rankin Scale, although not entirely focused on cognition, would be useful as well.

- study variables:

- time of CT scan should be precisely stated from stroke symptom onset: indeed, standardized timing of CT scan is fundamental for accurate volume assessment; characteristics of the brain infarct on CT scan evolve overtime in the acute and subacute phase of stroke, since the lesion progressively acquires better contour definition and hypodensity, as well as for the development and resolution of cerebral edema, etc.

- it is not clear which method has been used to calculate the infarct volume (pixel thresholding or manual tracing of the lesion perimeter or a combination of both?); moreover, other important features are missing: which machine and which software have been employed? which was the slice thickness and was it homogenous among the scans? how the lesion hypodensity has been distinguished from cerebrospinal fluid in ventricles or sulci and older infarcts? How was the lesion volume calculated in case of movement artifacts?

- imaging information about older silent infarcts, lacunes, multiple acute strokes, lesion side (right/left), extent of white matter disease, hemorrhagic transformation, lesion in anterior/posterior circulation is missing, however it is desirable in an imagin study of association with post-stroke cognitive performances.

- in the presented results, ischemic volume ranges from 0.02 to 1.49 mL, which seem to be super small lesions- please verify the numbers.

b. VEGF serum levels: the Authors found higher VEGF levels in patients with MOCA<26 at 3 months, although not significant (p=0.106). A multivariable analysis - showing the association of higher VEGF levels with poor post-stroke cognitive performance - was performed only after that a cut-off value was derived from ROC curves.

In the discussion authors overinterpret the association of VEGF levels with VCI with a causal relationship. Authors should rather point to a mere association. In the study it is not possible to ascertain causes and effects of VCI and VEGF levels and overinterpretations and causal relationship should be avoided. Extending the discussion to the association of VEGF levels with other stroke parameters (infarct size, stroke severity, stroke aetiology etc.) would add further insights. Indeed increased VEGF levels might be rather the consequence of increased stroke volume that per se can cause increased VCI.

- additional comments: it is not stated whether the study was retrospective or perspective and if the examiners were blinded; flow chart of the inclusion/exclusion process is missing; levels for a variable to be entered in the multivariable model have not been pre-specified; a careful revision of the general syntax is recommended.
---

## [Author Response · Author response to Decision Letter 0]

11 Aug 2020

Response to Reviewers

Reviewer 1:

Dear authors,

The major concern is that the severity of neurological impairment (National Institute of Health Stroke Scale) was not assessed and may be very different between groups with and without Vascular Cognitive Impairment (VCI and non-VCI). Therefore, the higher levels of Vascular Endothelial Growth Factor may not be associated exclusively with ICV. Another concern is whether or not the cognitive state previous to the study was accessed. The Montreal Cognitive Assessment-Indonesia version (MoCA-INA) score on day 5 is unclear for both groups. These are the main remarks that should be discussed.

Responses:

We thank reviewer for your valuable suggestions. Here are our responses for your review:

In our study, we also assessed NIHSS on day 5 (although previously we have not showed it yet in our manuscript). In our revised manuscript, table 1 shows that there was no different of NIHSS between post-stroke cognitive impairment (PSCI, previously wrote as VCI) and non-PSCI group. Both groups only had two categories of NIHSS which were minor and moderate. Moreover, we have carefully re-analyzed the data and added more analysis regarding the effect of NIHSS to VEGF. In table 3 (multivariate analysis), we show that higher VEGF was significantly associated with PSCI even after adjusting the NIHSS. In addition, interaction analysis showed no interaction effect between VEGF and NIHSS, as well as VEGF and infarct volume. This further supports that higher VEGF level was independently associated with PSCI.

We did not assess the cognitive state of the patients before the study began. However, as soon as the patients were admitted, we evaluated whether they had any symptoms of dementia or cognitive impairment by administering Short IQCODE (Short form of the Informant Questionnaire on Cognitive Decline in the Elderly) to the family or caregiver. Short IQCODE can be used as a screening tool for dementia and frequently used for exploring pre-existent cognitive deficits (1-5). Therefore, patients with dementia tendencies or any cognitive deficits before the index stroke were excluded in our study.

We did not analyze MoCA-INA on day 5 for both groups because the cognitive state during acute stroke is still fluctuated. In addition, the assessment of cognitive function to determine PSCI is mostly done on 3 months after stroke. This time period has been widely accepted in most researches for determining the term “PSCI” (6-9). Hence, we have amended the sentences about the cognitive assessments in our revised manuscript.

Further notes:

- Page three, line fifty-six: “In Indonesia, the prevalence of VCI after ischemic stroke is still high, reaching 68.2%”. It is necessary to add the ictus time (The ischemic stroke within 6 months).

Response: Thank you for your valuable correction. We have added the ictus time in our revised manuscript (“In Indonesia, the prevalence of VCI within 6 months of ischemic stroke is still high, reaching 68.2%”).

- Page three, line sixty-seven: “Another study found that the level of VEGF on day 5 of stroke onset could predict the outcome in stroke patients”. Is the level of VEGF higher or lower?

Response: We have amended the sentence above and put it in the next paragraph in the revised manuscript. It becomes: “A study found that lower VEGF level on day 5 of stroke onset was associated with brain edema in acute ischemic stroke patients [12]. Although they did not directly measure the cognitive function, the existing of brain edema itself could cause cognitive dysfunction.” Thank you for your suggestion.

- Were there patients with temporal lobe and/or hippocampus injuries? Such discussion has not been elaborated in the article.

Response: in our study there were no patients with temporal lobe or hippocampal injuries. Therefore, we did not provide any discussion for these cases. 

- Page sixteen, line two hundred and seventy-four: “Another study showed lower serum VEGF expression was found in ischemic stroke patients, but not in control group”. It is also important to mention the ictus time here.

Response: We have amended the sentence above in our revised manuscript. It becomes: “Furthermore, other study revealed that serum VEGF expression measured on day 5 of stroke onset was significantly lower in ischemic stroke patients than in control group”. 

 

Reviewer 2:

In the manuscript entitled “Higher level of acute serum VEGF and larger infarct volume are associated with higher risk of vascular cognitive impairment after ischemic stroke” the Authors examined a population of 56 ischemic stroke patients and investigated the possible associations of serum levels of the Vascular Endothelial Grow Factor (VEGF) and ischemic lesion volume with vascular cognitive impairment at 3 months from stroke onset.

VEGF indeed is an important growth factor mediating several protective but also some detrimental effect in the ischemic brain.

Following aspects should be considered to improve the manuscript:

The possible inter-relationship between the two study variables (serum VEGF and lesion size) has not been investigated, so that the two variable appear poorly integrated and the manuscript loses cohesiveness.

Response: We have carefully re-analyzed the data and added interaction analysis between VEGF and lesion size (infarct volume) in our revised manuscript. However, after adjusting with the covariates, there was no significant interaction effect between serum VEGF level and infarct volume. Therefore, we suggest that higher VEGF level was independently associated with PSCI at 3 months after stroke onset. We have also revised the discussion to make the sentences more integrated and coherent. Thank you for your valuable suggestions.

- endpoint of the study: The endpoint of the study is the development of vascular cognitive impairment at 3 months from stroke onset. The Authors evaluated the development of vascular cognitive impairment by means of the dichotomized Montreal Cognitive Assessment (MoCA) scale (with a cut-off of 26).

- Since all included patients suffered from ischemic stroke, the term “post-stroke cognitive impairment” instead of “vascular cognitive impairment” appears more accurate and is suggested to the Authors (https://doi.org/10.1186/s12916-017-0779-7).

Response: Thank you for your correction. We agree with your suggestion. In our revised manuscript, we have replaced the term VCI with PSCI.

- Various tools are available to assess cognition after stroke, with clear standards missing, and need to be discussed. Impact on the activities of daily living, perceived impairment, as well as subdomain scores of the MoCA have not been investigated, but would provide additional information.

Response: The reason why we used MoCA to define PSCI is because MoCA is more sensitive to detect MCI and dementia in stroke patients than MMSE. Indeed, we also measured the activities of daily living (ADL) and the symptoms of depression using Hamilton depression rating scale for all the subjects. We only included subject with PSCI no dementia. Therefore, subjects with any impairment found in ADL were excluded. In addition, subjects with any symptoms of depression were also excluded due to interference with cognitive function. Moreover, subjects with aphasia, dysphasia, visual and hearing impairment, illiterate, and dysarthria were also excluded. We have added this information in study design.

We have also added the analysis of association between higher VEGF level with cognitive function (each domain from MoCA-INA) in table 4. We found that higher VEGF level was significantly associated with deficits in visuospatial and recall domains for PSCI patient. However, mechanism underlying this result still needs further investigation.

- Importantly, it has not been clarified how the test has been administered in cases of stroke patients with aphasia, dysphasia, dysarthria, hearing impairment, inability to use the dominant arm, or visual impairment.

Response: As we mention above, we excluded the patients with aphasia, dysphasia, dysarthria, hearing and visual impairment. Patients with inability to use the dominant arm were told to use non-dominant arm for performing the test. Scoring of MoCA-INA is based on a guideline issued by Indonesian Neurological Association. The guideline includes how the test is administered and scored. Each test item has its own criteria for scoring purposes. We have added this information in Cognitive assessments section in our revised manuscript.

- Similarly, stroke severity, lesion side (right/left), acute stroke treatment are easily available data with impact on stroke outcome that should have been included in patient description, as well as their association with cognitive performances should have been explored.

Response: We thank the reviewer for your valuable suggestion. Indeed, we have the data of NIHSS, lesion side, and acute stroke treatment in our study. We have added stroke severity (as measured by NIHSS) and lesion side and included them into the analysis in our revised manuscript. For acute stroke treatment, due to all patients received similar standard treatment based on the stroke guideline, we did not include this variable into the analysis. However, we explored the impact of anti-dyslipidemic agents because the drugs were only given to patients with dyslipidemia.

- Whereas the Authors stated that patients with history of dementia have been excluded from the sample, it is fundamental to clarify how it was assessed: e.g. by reports of an established diagnosis, screening questionnaire administered to the patient or to a family member, etc. An easy assessment of pre-stroke functional status, such as modified Rankin Scale, although not entirely focused on cognition, would be useful as well.

Response: We excluded patients with history of dementia by using Short IQCODE (Short form of the Informant Questionnaire on Cognitive Decline in the Elderly). This is a reliable questionnaire to detect pre-existing cognitive impairment. This questionnaire is administered to the family or caregiver who have closely observed the patients’ behavior over the 5 years prior to the disease (1-5).

- study variables:

- time of CT scan should be precisely stated from stroke symptom onset: indeed, standardized timing of CT scan is fundamental for accurate volume assessment; characteristics of the brain infarct on CT scan evolve overtime in the acute and subacute phase of stroke, since the lesion progressively acquires better contour definition and hypodensity, as well as for the development and resolution of cerebral edema, etc.

Response: Time to CT scan was immediately after patients’ admission in the emergency room. All the patients included in our study had stroke onset less than 5 days. Therefore, the time of CT scan was varied between immediately on day 0 until day 4 of stroke onset. Unfortunately, we did not record the time of CT scan precisely. We will put this issue in our limitation. Thank you for your valuable suggestion.

- it is not clear which method has been used to calculate the infarct volume (pixel thresholding or manual tracing of the lesion perimeter or a combination of both?); moreover, other important features are missing: which machine and which software have been employed? which was the slice thickness and was it homogenous among the scans? how the lesion hypodensity has been distinguished from cerebrospinal fluid in ventricles or sulci and older infarcts? How was the lesion volume calculated in case of movement artifacts?

Response: We have added more clear information about this issue in our revised manuscript. Thank you for your suggestion. 

- imaging information about older silent infarcts, lacunes, multiple acute strokes, lesion side (right/left), extent of white matter disease, hemorrhagic transformation, lesion in anterior/posterior circulation is missing, however it is desirable in an imagin study of association with post-stroke cognitive performances.

Response: We have added more clear information about this issue in our revised manuscript. Thank you for your suggestion. 

- in the presented results, ischemic volume ranges from 0.02 to 1.49 mL, which seem to be super small lesions- please verify the numbers.

Response: Philips IntelliSpace Portal version 10.1 software were used for hypodense area segmentation. Segmentation module and volume rendering technique were used to obtain 3D visualization of the lesion in each segmentation. Infarct volume calculation was done manually using hand tracing method at the edge of hypodense area by expert radiologist specialize in brain imaging. Edge of lesion segmentation was carefully examined, slide by slide, until the last slide in which the minimum hypodense lesion area was still visualized. The segmentation process was done axially which then confirmed by the coronal and sagittal planes. It is true that the infarct volume was converted from mm3 to ml. It ranges from 20 mm3 to 1490 mm3 or 0.02 ml to 1.49 ml.

b. VEGF serum levels: the authors found higher VEGF levels in patients with MOCA<26 at 3 months, although not significant (p=0.106). A multivariable analysis - showing the association of higher VEGF levels with poor post-stroke cognitive performance - was performed only after that a cut-off value was derived from ROC curves.

In the discussion authors overinterpret the association of VEGF levels with VCI with a causal relationship. Authors should rather point to a mere association. In the study it is not possible to ascertain causes and effects of VCI and VEGF levels and overinterpretations and causal relationship should be avoided. Extending the discussion to the association of VEGF levels with other stroke parameters (infarct size, stroke severity, stroke aetiology etc.) would add further insights. Indeed increased VEGF levels might be rather the consequence of increased stroke volume that per se can cause increased VCI.

Response: Thank you for your valuable suggestions. We have amended our statement about the association of VEGF with PSCI in discussion section of our revised manuscript. We have also carefully re-analyzed the data and included other variables such as NIHSS and infarct side. After careful re-analyzing and adjusting with covariates, we found that higher VEGF level and larger infarct volume were more frequently associated with PSCI. We suggest that those two parameters could predict the development of PSCI 3 months after index stroke. Furthermore, no significant interaction effect between VEGF and infarct volume, as well as VEGF and NIHSS towards PSCI.

- additional comments: it is not stated whether the study was retrospective or perspective and if the examiners were blinded; flow chart of the inclusion/exclusion process is missing; levels for a variable to be entered in the multivariable model have not been pre-specified; a careful revision of the general syntax is recommended.

Response: This study was prospective. Patients who consented to the study were prospectively enrolled and followed up at 3 months after stroke onset. All the examiners were blinded. We have added this information in study design. 

We have provided a flow chart of the inclusion/exclusion process in our revised manuscript. We have also added some information about variables used in the multivariate model in methods section. 

1. Jorm A, Christensen H, Henderson A, Jacomb P, Korten A, Mackinnon A. Informant ratings of cognitive decline of elderly people: relationship to longitudinal change on cognitive tests. Age and Ageing. 1996;25(2):125-9.

2. Jorm AF. The Informant Questionnaire on cognitive decline in the elderly (IQCODE): a review. International psychogeriatrics. 2004;16(3):275.

3. De Jonghe J, Schmand B, Ooms M, Ribbe M. Abbreviated form of the Informant Questionnaire on cognitive decline in the elderly. Tijdschrift voor gerontologie en geriatrie. 1997;28(5):224-9.

4. Morales J-M, Gonzalez-Montalvo J-I, Bermejo F, Del-Ser T. The screening of mild dementia with a shortened Spanish version of the “Informant Questionnaire on Cognitive Decline in the Elderly”. Alzheimer Disease & Associated Disorders. 1995;9(2):105-11.

5. Perroco TR, Damin AE, Frota NA, Silva M-NM, Rossi V, Nitrini R, et al. Short IQCODE as a screening tool for MCI and dementia: preliminary results. Dementia & Neuropsychologia. 2008;2(4):300-4.

6. Sun J-H, Tan L, Yu J-T. Post-stroke cognitive impairment: epidemiology, mechanisms and management. Annals of translational medicine. 2014;2(8).

7. Douiri A, Rudd AG, Wolfe CD. Prevalence of poststroke cognitive impairment: South London stroke register 1995–2010. Stroke. 2013;44(1):138-45.

8. Cumming T, Churilov L, Lindén T, Bernhardt J. Montreal Cognitive Assessment and Mini–Mental State Examination are both valid cognitive tools in stroke. Acta Neurologica Scandinavica. 2013;128(2):122-9.

9. Melkas S, Jokinen H, Hietanen M, Erkinjuntti T. Poststroke cognitive impairment and dementia: prevalence, diagnosis, and treatment. Degenerative neurological and neuromuscular disease. 2014;4:21-7.

---

## [Decision Letter · Decision Letter 1]

7 Sep 2020

Higher level of acute serum VEGF and larger infarct volume are more frequently associated with post-stroke cognitive impairment

PONE-D-20-13907R1

Dear Dr. Vidyanti,

We’re pleased to inform you that your manuscript has been judged scientifically suitable for publication and will be formally accepted for publication once it meets all outstanding technical requirements.

Kind regards,

Marietta Zille, PhD

Academic Editor

PLOS ONE

Additional Editor Comments (optional):

Some more grammatical corrections to be made:

Abstract:

• Line 33: It should be: “We performed a ROC curve analysis..:”

• P values should be written p=…

• Line 47: “and thus”

Introduction:

• Line 56: “Based on the…”

• Line 57: “at 5 years after stroke”

• Line 68: replace “could” by “may”

• Line 70: remove “attack”

• Line 73: remove “the existing of“ and replace “could” by “may”

• Line 74: “in vitro” should be in italics

• Line 76: it should be “models”

• Line 82: replace “proved“ by “suggest“ and “might” by “may”

• Line 84/85: “which is associated”

• Line 92: replace “knowing“ by “understanding“

• Line 94: replace “made” by “developed”

Methods:

• Line 106: replace “have” by “having”

• Line 120: “83 patients that were“

• Line 121: replace “could be” by “was”

• Line 123: “samples”

• Section on “Analysis of VEGF” (lines 171-183) should be written in past tense.

• Line 173: Include the catalogue number of the VEGF antibody.

• Line 178: “wash buffer”

• Line 179: “human”

• Line 180/181: “substrate solution”

• Line 182: “stop solution”

• Line 190: “use the non-dominant”

• Line 196: replace “thus” by “which”

Results:

• Write p values with p=…

• Line 226: remove “There were”

• Line 229: “differences”

• Line 229-231: Remove “The VEGF level was higher in PSCI group than the counterpart group although it did not reach statistically significance.”

• Line 247/248: “Fig 2. ROC curve for the discrimination quality of VEGF level in PSCI patients (weak categorization). Serum level of VEGF at cut-off point of 519.8 pg/ml with AUC 0.630 were used.”

• Line 250/251: “Fig 3. ROC curve for the discrimination quality of infarct volume in PSCI patients (moderate categorization). Infarct volume at cut-off point of 0.054 ml with AUC 0.739 were used.”

Discussion:

• Line 306: remove “could”

• Line 309: replace “the” by “a”

• Line 310: “had a protective effect”

• Line 312: “contributes”

• Line 315: replace “might” by “may”

• Line 318: italize “in vitro” and “in vivo”

• Line 319: replace “could improve” by “improved”

• Line 326: “with a prior study”

• Line 346: replace “could” by “may”

• Line 349, 361, 368, 372, 373, 390: replace “might” by “may”

Reviewers' comments:

Reviewer's Responses to Questions

6. Review Comments to the Author

Reviewer #1: I recommend approval of the manuscript entitled "Higher level of acute serum VEGF and larger infarct volume are more frequently associated with post-stroke cognitive impairment" in its current form for publication. Very interesting paper and the modifications were adequate.

Reviewer #2: thank You for the detailed responses to the reviewers requests.

all questions have been answered. the title might be shortened.

proposal "Increased serum VEGF and infarct volume are associated with post-stroke cognitive impairment"

---

## [Editor Report · Acceptance letter]

22 Sep 2020

PONE-D-20-13907R1 

Higher level of acute serum VEGF and larger infarct volume are more frequently associated with post-stroke cognitive impairment 

Dear Dr. Vidyanti:

I'm pleased to inform you that your manuscript has been deemed suitable for publication in PLOS ONE. Congratulations! Your manuscript is now with our production department. 

Kind regards, 

on behalf of

Dr. Marietta Zille 

Academic Editor

PLOS ONE